# New Insight into Muscle-Type Cofilin (CFL2) as an Essential Mediator in Promoting Myogenic Differentiation in Cattle

**DOI:** 10.3390/bioengineering9120729

**Published:** 2022-11-25

**Authors:** Yujia Sun, Tianqi Zhao, Yaoyao Ma, Xinyi Wu, Yongjiang Mao, Zhangping Yang, Hong Chen

**Affiliations:** 1Joint International Research Laboratory of Agriculture and Agri-Product Safety, The Ministry of Education of China, Institutes of Agricultural Science and Technology Development, Yangzhou University, Yangzhou 225009, China; 2Key Laboratory of Animal Genetics, Breeding and Reproduction of Shaanxi Province, College of Animal Science and Technology, Northwest A&F University, Xianyang 712100, China; 3Key Laboratory of Animal Genetics & Breeding and Molecular Design of Jiangsu Province, Yangzhou University, Yangzhou 225009, China; 4College of Animal Science, Xinjiang Agricultural University, Urumqi 830052, China

**Keywords:** muscle-type CFL2, spatiotemporal expression, bta-miR-183, DNA methylation, cattle, myoblasts differentiation

## Abstract

Meat quality and meat composition are not separated from the influences of animal genetic improvement systems; the growth and development of skeletal muscle are the primary factors in agricultural meat production and meat quality. Though the muscle-type cofilin (CFL2) gene has a crucial influence on skeletal muscle fibers and other related functions, the epigenetic modification mechanism of the *CFL2* gene regulating meat quality remains elusive. After exploring the spatiotemporal expression data of *CFL2* gene in a group of samples from fetal bovine, calf, and adult cattle, we found that the level of *CFL2* gene in muscle tissues increased obviously with cattle age, whereas DNA methylation levels of *CFL2* gene in muscle tissues decreased significantly along with cattle age by BSP and COBRA, although DNA methylation levels and mRNA expression levels basically showed an opposite trend. In cell experiments, we found that bta-miR-183 could suppress primary bovine myoblast differentiation by negatively regulated CFL2. In addition, we packaged recombinant adenovirus vectors for *CFL2* gene knockout and overexpression and found that the *CFL2* gene could promote the differentiation of primary bovine myoblasts by regulating marker genes *MYOD*, *MYOG* and *MYH3*. Therefore, CFL2 is an essential mediator for promoting myogenic differentiation by regulating myogenic marker genes in cattle myoblasts.

## 1. Introduction

Meat quality and meat composition are not separated from the influences of animal genetic improvement systems. Previous studies have indicated that meat quality is tightly correlated with muscle composition and structure and the histological properties of muscle fibers [1,2]. It is therefore crucial to further study the molecular genetic mechanism of skeletal muscle growth and development [3]. Moreover, one study examined cofilin gene function as a key regulator of actin assembly on myoblast proliferation and differentiation [4]. In healthy mouse muscle cells, the type of cofilin from non-muscle cofilin transits to muscle cofilin [5]. In our previous study, we found that CFL1-mediated epigenetic regulation mechanisms were involved in the myogenic growth and differentiation [6]. Muscle-type cofilin is a new member of the cofilin protein family, which mainly regulates the expression of skeleton muscle and cardiac muscle. In mammals, muscle-type cofilin is also named CFL2 (cofilin-2, M-cofilin). Studies have shown that CFL2 was mainly detected in adult skeletal muscle and performs an essential role in skeleton muscle development and maintenance [7,8,9].

As the only subtype of mature skeletal muscle, CFL2 also performs a crucial function in maintaining the constitution of muscle fibers. A previous report showed that CFL2 is the primary expressed isoform of cofilin in adult muscle of human and murine, especially in skeletal and cardiac muscles [10]. CFL2 is essential for myoblast proliferation and differentiation by modulating the expression of myogenic transcription factors in C2C12 cells. Mai presented an unusual regulatory pattern, wherein CFL2 mediates myoblasts differentiation in C2C12 cells, significantly, CFL2 depletion suppressed cell differentiation, accelerated cell proliferation and induced cell cycle from the G0/G1 stages to the G2/M stages [11]. Considering *CFL2* a candidate gene for nemaline myopathy, some studies have shown that the mutation from G to A in the coding region position 103 of the *CFL2* gene can induce the occurrence of human nemaline myopathy, and the *CFL2* gene can regulate the expression of key factors CAM and MEF2C in myofibroblast signal pathway, resulting in a change in muscle properties [12]. A *CFL2*^A35T/A35T^ knock-in mouse model indicated that the expression levels of *CFL2* mRNA and full-length transcript decreased significantly in skeletal muscles and muscle biopsy samples of a p.A35T mutation patient, respectively [13]. Sun examined the association of the *CFL2* gene polymorphisms with performance traits in Qinchuan (QC) cattle. Base mutations at three single nucleotide polymorphisms sites were found in the coding region, which can significantly affect body length and body mass of cattle, indicating that the *CFL2* gene can be used as a genetic marker for growth traits in cattle breeding program [14].

According to the regulation of the *CFL2* gene on skeletal muscle fibers and other related functions, it can be observed that the *CFL2* gene has an important influence on the growth and development of myocytes. However, if we want to study the way in which it affects skeletal muscle and the specific functional mechanism of muscle, we need to further explore the epigenetic regulation mechanism of *CFL2* gene. Epigenetics has been recently undergoing the evolution of research field from diverse and complicated phenomena to profound and defined field [15]. Here, we investigated the regulatory functions of CFL2 from the angle of spatiotemporal expression pattern, DNA methylation and miRNA–target relationship. Our findings have made comprehensive analysis of the molecular regulation mechanisms of *CFL2* gene on myogenesis. This has profound significance in understanding the complicated biological characteristics of meat quality and will provide a very strong basis for theoretical research for the subsequent genetic improvement of cattle breeds.

## 2. Materials and Methods

### 2.1. Tissue Collection and Cell Culturing

Seven tissue samples (heart, liver, spleen, lung, kidney, fat and *longissimus dorsi* muscle tissues) were harvested from three critical phases of muscle formation and maturation: 90-day embryonic (FB, fetal bovine, n = 3), 1-month postnatal (calf, n = 3), and 24-month-old (AC, adult cattle, n = 3) samples. All tissue samples used in this study were obtained from a local QC cattle breeding center (Xi’an, China); they were all female cattle and were raised in similar conditions.

The HEK293, 293A and C2C12 cell lines were the preservative cell lines from American ATCC in our laboratory. Primary bovine myoblasts (PBMs) were harvested from fetal bovine *longissimus dorsi* muscle and cultured by collagenase-I digestion as previously described [13]. Cell lines and PBMs were cultured following our previously established protocols [6].

### 2.2. Overexpression or Knock-Down Vector Construction of CFL2

*CFL2* overexpression primers containing restriction sites *Kpn*I (CFL2-CMV-F) and *Hin*dIII (CFL2-CMV- R) (TaKaRa, Dalian, China) were designed and synthesized. *CFL2* knock-down primers containing restriction sites *Bam*HI (shCFL2-1F, shCFL2-2F) and *Hin*dIII (shCFL2-1R, shCFL2-2R) (TaKaRa, Dalian, China) were designed and synthesized by BlockiT shRNA interference system. All primer information containing negative control (shRNA-NC-F/R, NC) is displayed in Table 1. Adenovirus vectors of pAdEasy-1/pAdtrack-CMV-CFL2 (CFL2-CMV) and pAD/PL-DEST/CMV-GFP/U6-shCFL2 (shCFL2-1 and shCFL2-2) were successfully constructed, and adenovirus recombinants displayed biological activity (Appendix A). The subsequent process for packaging recombinant adenovirus followed a previous protocol [16].

### 2.3. Cell Treatment and Cell Differentiation

For the biochemical study, differentiation medium (DM, 2% horse serum) replaced growth medium (GM, 10% fetal bovine serum) after the cell density reached 70~80%. C2C12 cells were cultured for 1, 2, 4, 6 and 8 days. In the meantime, plasmid CFL2-CMV, plasmid shCFL2-1, pAdTrack-CMV vector (control group) or siRNA negative control vector (NC) were injected into PBMs by lipofectamine^TM^ 2000 (Invitrogen, Carlsbad, CA, USA) for myoblasts differentiation. PBMs were harvested at 1, 3, 5 and 7 days. HEK293 cells were transfected for 48 h to check the expression efficiency of recombinant adenovirus. Bta-miR-183 mimic (overexpression) or inhibitor (interference) was injected into PBMs and induced differentiation for 4 d.

### 2.4. Dual Luciferase Reporter Assay

The targeting sites between the *CFL2* gene and bta-miR-183 were amplified using CFL2-wild-F/R primers (Table 2). An eight-base deletion of *CFL2* 3′UTR in the bta-miR-183 binding site was generated with mutagenic primers CFL2-mut-F/R (Table 2). psiCHECK-2 vector (Promega, Madison, WI, USA) wtih restriction enzymes *Xho*I and *Not*I (TaKaRa, Dalian, China) was used to produce psiCHECK-2-CFL2-wild or psiCHECK-2-CFL2-mutate (CFL2-Luc or CFL2-del Luc). Bta-miR-183 mimic and inhibitor were purchased from GenePharma (Shanghai, China). The dual-luciferase activity was analyzed by the Dual-Luciferase Reporter (DLR) Assay System (Promega, Madison, WI, USA) according to the manufacturer’s instructions.

### 2.5. Quantitative Real-Time PCR and Western Blot Assay

Total tissues or cellular RNA were obtained using TRIzol reagent (TaKaRa, Dalian, China), cDNA was synthesized using PrimeScript RT reagent Kit (TaKaRa, Dalian, China), and qRT-PCR was performed using SYBR Green Master Mix Reagen kit (GenStar, Beijing, China). U6 or GAPDH were synthesized to normalize the mRNA or protein level. All primers used for quantitative real-time PCR (qRT-PCR) are listed in Table 3.

The Western blot (WB) experiment was examined as Appendix A. The primary antibodies anti-CFL2 (ab14134) and anti-GAPDH (ab9485) were purchased from Abcam (Cambridge, UK), secondary antibody anti-immune rabbit IgG-HRP (LK2001) was purchased from Sungene Bio (Tianjin, China) and ECL luminous fluid (Solarbio, Beijing, China) was used to detect the antibody reacting bands.

### 2.6. Bisulfite Sequencing Polymerase Chain Reaction and Combined Bisulfite Restriction Analysis

*Longissimus dorsi* DNA were harvested from 90-day embryonic (4 male fetal bovine, FB) and 24-month-old (4 male adult cattle, AB) tissues by methylSEQr Bisulfite Conversion Kit (Applied Biosystems, Foster City, CA, USA) according to the manufacturer’s instructions. The Bisulfite Sequencing Polymerase Chain Reaction (BSP) primers from the *CFL2* differentially methylated region (DMR) were designed by online software MethPrimer [17] (Table 4). All experiments performed 15 cloning sets (sequenced 5 clones, n = 3).

The combined bisulfite restriction analysis (COBRA) was performed as Appendix A. We performed COBRA analysis (on the same batch PCR products which were treated for BSP sequencing) using restriction enzymes *Hin*fI (“GANTC”) for CFL2 DMR.

### 2.7. Statistical Data Analysis

All data were presented as the mean ± SE (standard error). The outcomes were resolved by the 2^−ΔΔCt^ method [18] and assessed by SPSS statistics V18.0. One-way ANOVA was performed for multiple comparisons using GraphPad Prism V8.0 (GraphPad Software, San Diego, CA, USA). *p*-value < 0.05 or <0.01 were judged as statistical differences or significant differences, respectively.

BSP data were analyzed by online software QUMA [19] and BiQ Analyzer [20]. The differences of methylation levels were analyzed by SPSS statistics V18.0.

## 3. Results

### 3.1. Spatiotemporal Expression Profiles of CFL2 Gene in Different Tissues

In order to investigate the *CFL2* role in cattle, our study first detected the mRNA expression levels of *CFL2* in seven different tissues of three developmental phases in QC cattle using qRT-PCR. The expressions of the *CFL2* gene have the same performance in fetal bovine, calf and adult cattle. *CFL2* gene showed the highest expression in muscle tissues, followed by expression in heart and lowest expression in spleen in the three growth and development stages (*p* < 0.05) (Figure 1A–C). At the fetal bovine stage, the mRNA levels of *CFL2* gene among liver, kidney and lung tissues were not significantly different (*p* > 0.05) (Figure 1A). At the calf stage, the mRNA levels of *CFL2* gene in fat and liver were higher than in kidney and lung (*p* < 0.05) (Figure 1B), whereas at the adult cattle stage, the mRNA levels of the *CFL2* gene among the lung, kidney and fat tissues were not statistically different (*p* > 0.05) (Figure 1C).

We further detected the *CFL2* gene mRNA expression level in the same three tissue growth and development stages of QC cattle; the expression trends are shown in Figure 1D. In heart, liver and lung tissues, the expression of *CFL2* gene indicated an upward trend in three growth and development stages along with cattle age. Additionally, due to no fat deposition in FB stage, adipose tissues were observed in calf stage and AC stage, and the results revealed that mRNA expression levels of *CFL2* gene have an extremely significant increase in adult bovine compared to that of the calf (*p* < 0.01). In the spleen, the *CFL2* gene has a higher expression level in the calf; the mRNA level was unexpectedly upregulated in the calf relative to that of the FB and AC stages (*p* < 0.05), whereas the mRNA expression level in the AC period presented a diminishing drift compared to those of the FB stage and calf stage in kidney tissues (*p* < 0.01), and the differences in mRNA expression levels in the fetal bovine and calf period were insignificant (*p* > 0.05). In muscle tissue, the *CFL2* gene mRNA level presented an extremely significantly upward trend with cattle age (*p* < 0.01) (Figure 1E).

### 3.2. DNA Methylation Analysis of CFL2 Promoter Region in Muscle Tissues

We explored DNA methylation assays on myogenesis and muscle maturation to demonstrate the impression of *CFL2* gene differential methylation level on myoblast differentiation at the FB and AC stages. The CpG islands seated in *CFL2* DMR were predicted by MethPrimer [17]; one CpG island of *CFL2* gene promoter region was taken as research objective (Figure 2A). We found 8 CpGs at *CFL2* DMR (177 bp) by BSP-amplified sequences. Every sample sequenced 15 clones; then, we detected 120 CpGs in every muscle sample at *CFL2* DMR. The methylated CpGs percentages of *CFL2* gene were 61.7%, 61.7%, 60.8% and 60.0% in the FB group; there were 47.5%, 47.5%, 46.7% and 48.3% in the AC group. Statistical results presented that the FB group (mean 61.1%) had prominently higher methylation levels than the AC group (mean 47.5%) (Figure 2B). In order to improve the specificity and to decrease the false positives of BSP sequencing results, we further detected COBRA assay by using restriction enzymes *Hin*fI (“GANTC”) (Figure 2C). At the fourth and fifth CpG sites, 177 bp PCR fragments of *CFL2* DMR were digested with *Hin*fI and obtained 177, 128 and 49 bp fragment bands by COBRA (Figure 2D). The outcomes were in accordance with the results of BSP sequencing.

In cells, methyl moieties may directly regulate the action of transcription factors. To further testify the relationship between DNA methylation level and the expression level of *CFL2* gene in muscle tissues at the FB group and the AC group, the following experiment tested the mRNA level of the *CFL2* gene. In contrasted with the FB group, the mRNA level of the *CFL2* gene in the AC group was statistically higher (*p* < 0.05) (Figure 2E).

### 3.3. bta-miR-183 Target Muscle Type CFL2

In addition to influences on muscle, previous study showed that bta-miR-183 represented regulator of proteolysis in muscle, which affects meat tenderness. To verify this hypothesis in C2C12 cell lines, after treating bta-miR-183, mimic or inhibitor, the mRNA level was statistically induced or suppressed (Figure 3A). After performing induction for 4 days for cell differentiation, we found that the mRNA levels of myoblast differentiation marker genes *MYOD*, *MYOG* and *MYH3* were downregulated by bta-miR-183 mimic (*p* < 0.05). Interestingly, the mRNA levels of these three marker genes were statistically upregulated by the bta-miR-183 inhibitor (*p* < 0.01) (Figure 3B,C). These data revealed that bta-miR-183 could suppress PBM differentiation.

To further determine the effect of *CFL2* gene on myoblast differentiation, we examined whether bta-miR-183 directly targets and regulates *CFL2* expression. As shown in Figure 3D,E, the mRNA and protein levels of *CFL2* were prominently upregulated or downregulated after being treated with bta-miR-183 inhibitor or mimicked in PBMs (*p* < 0.01 or *p* < 0.05). These results indicated that bta-miR-183 negatively regulated the expression of *CFL2*. Using the in silico method, we predicted a potential binding site for the bta-miR-183 seed sequence in 3′UTR of *CFL2* mRNA in ten different species, which revealed highly conservative bta-miR-183 target sites (Figure 3F). To verify direct interaction sites between bta-miR-183 and *CFL2*, we generated a luciferase reporter psiCHECK-2 construct containing a *CFL2* 3′UTR segment or delete binding site for bta-miR-183 (Figure 3G). Compared to only containing the NC vector (*p* < 0.01), renilla luciferase activity was prominently reduced when bta-miR-183 mimic and CFL2-Luc co-transfected PBMs, while fluorescent reactive was resumed after bta-miR-183 mimic co-transfected with CFL2-del Luc (*p* > 0.05) (Figure 3H). The base data showed that miR-183 directly binds to *CFL2* 3′-UTR.

### 3.4. Effects of CFL2 Gene on Myoblasts Differentiation

To further investigate whether *CFL2* influences the expressions of myogenic factors, we generated an adenovirus overexpression and interference construct (Appendix A). mRNA and protein assays demonstrated that *CFL2* expression levels were obviously up-regulated or down-regulated after transfection of CFL2-CMV or shCFL2 in HEK293 cells (*p* < 0.01); meanwhile, shCFL2-1 had stronger knock-down effects than shCFL2-2 (Figure 4A–C). After transfecting CFL2-CMV or shCFL2-1 into C2C12 cells, the marker genes *MYH3*, *MYOG* and *MYOD* were assessed on differentiation day 4. We found the mRNA levels of *MYOD* and *MYH3* were upregulated after being transfected with CFL2-CMV (*p* < 0.05), while *MYOG* level was not apparently changed (*p* > 0.05). After transfecting shCFL2-1, the mRNA expression levels of marker genes were extremely significantly decreased (*p* < 0.01) (Figure 4D,E). After being induced for 8 days (DM 1–8 d) in C2C12 cells, *CFL2* and marker genes of mRNA levels presented upward trends. During this process, the expression level of the *MYH3* gene was always lower than that of other genes, and *MYOD* gene continuously maintained a high expression level, except it was lower than *MYOG* gene on DM2. The *CFL2* gene and the *MYOD* gene displayed the identical differentiation trends with a downward trend during 6–8 days (DM 6–8 d) in the late differentiation stage (Figure 4F).

In vitro, PBM model is the optimal one for studying myoblast differentiation (Appendix A). After transfection with CFL2-CMV, the mRNA level of *CFL2* was significantly up-regulated at 3–7 days (*p* < 0.05 or *p* < 0.01) (Figure 5A,I) and the mRNA levels of *MYOD*, *MYOG* and *MYH3* were extremely significantly increased (*p* < 0.01), except that there was no obvious difference in *MYH3* at 1 day (*p* > 0.05) (Figure 5B–D). After transfection with shCFL2-1, the expression of *CFL2* was significantly decreased compared to control group (*p* < 0.05 or *p* < 0.01) (Figure 5E), and *MYOG* was significantly down-regulated during myoblast differentiation (*p* < 0.05 or *p* < 0.01) (Figure 5G,J). Meanwhile, the mRNA levels of *MYOD* and *MYH3* showed an obvious decrease after 3 days of myoblast differentiation (*p* < 0.05 or *p* < 0.01) (Figure 5F,H).

After transfection with *CFL2*-CMV, the expression trend of *CFL2* was gradually up-regulated during myoblast differentiation (DM 1–7 d) (*p* < 0.01); meanwhile, *MYOD* and *MYOG* showed a significantly upward trend, and *MYH3* slightly increased during myoblast differentiation (Figure 5I). After transfection with shCFL2-1, the expression of *CFL2* presented a significantly downregulated trend during myoblast differentiation (DM 1–7 d); in the meantime, the expression trends of *MYOD*, *MYOG* and *MYH3* were significantly downward (*p* < 0.05 or *p* < 0.01), and the peak values of *CFL2* and *MYH3* both occurred at day 5 after induction and then showed a slight increase after induction (Figure 5J). Our findings have made comprehensive analysis of the molecular genetic regulation mechanisms of CFL2 gene on myogenesis.

## 4. Discussion

The PBM model acts as an optimal model for perusing myoblast differentiation. It is primarily associated with muscle growth and development and can improve muscle quality [21]. During culturing, a change in the serum could stimulate myoblasts to differentiate to myocytes or myotubes; in the meantime, the expression of some genes regulated the phenotypes of meat quality by some special epigenetic mechanisms [22]. Our previous research showed that the *CFL2* gene was significantly associated with body mass in bovinae [14], and there is a strong association between skeletal muscle mass, muscle fiber characteristics, meat quality and meat yield of beef cattle. Consequently, it is especially crucial to further expound the epigenetic regulatory network of muscle-related genes.

As a muscle subtype, the *CFL2* gene was generally considered to be mainly expressed in mammalian muscle tissue. Our present study reveals the myogenic regulatory characteristics of the *CFL2* gene in myoblast differentiation. An analysis of the spatiotemporal expression showed that the *CFL2* gene is widely distributed in various organizations. This indicates that the *CFL2* gene has a wide range of biological functions, whereas at the three growth and developmental stages, it revealed a vary expression trend in seven tissues. Especially in AC group muscle tissues, the mRNA expression level of the *CFL2* gene was increased probably eight times in comparison with the FB group and showed a high expression pattern and a meaningful increasing trend along with the muscle growth and development. These findings seem to be in accordance with earlier research on mice muscle tissue [8] and further demonstrate that the cofilin expression subtype could gradually transition to M-CFL2 along with the growth and development of myocytes [5,23].

Skeletal muscle development can be split into two stages: the prenatal stage that affects the number of muscle fibers and the postnatal stage that generates the size of muscle fibers [24]. The expression change in *CFL2* in muscle growth was likely correlated with the regulatory pattern of external epigenetics. Generally, DNA methylation participates in suppression patterns during muscle growth and development in mammals by regulating gene expression. Previous study examined the DNA methylation and mRNA expression of the *SERPINA3* gene DMR in muscle tissues and demonstrated that mRNA expression levels of genes could be affected by DNA methylation levels considerably [25], and research has demonstrated that a number of methylated genes could act as candidate beef tenderness biomarkers [26]. In our prior study, there was a negative correlation between DNA methylation status and the expression level of the *IGF2* gene in muscle tissues during the FB and AC stages [27]. Compared to the FB group, the *CFL2* gene showed an obviously high mRNA expression level and relatively low DNA methylation extent in the AC group in this experiment; here, a detectable trend for mammalian DNA methylation patterns to vary in time and space was observed. In FB and AC, the question arises of how DNA methylated patterns arise in development and maintenance, and how they affect the genetic expression in the whole genome. Though these burning questions cannot be elucidated explicitly now, additional studies have addressed distinct hypotheses experimentally in muscle growth and development [28].

Although miRNAs act as an important factor in myogenesis and pathogenesis of muscle wasting, the regulatory mechanisms of miRNAs are mainly expressed by targeting mRNAs [4,29,30]. There are four miRNAs that have been experimentally validated to be capable of regulating bovine skeletal muscle development by miRNA–mRNA interactions; nine miRNAs have been identified modulating PBMs differentiation through binding functional genes experimentally [31] (Appendix A). Owing to the innumerable target genes, the epigenetic mechanisms of miR-183 seemed very complicated and even inconsistent during skeletal muscle differentiation. Several studies have expounded the miR-183 function on the apoptosis, proliferation and invasion of cancers by negatively regulated FOXO1 [32,33,34]. Our previous data exhibited that the *FOXO1* gene plays a major part in the bovine myoblast differentiation [22]. Thus, the function of miR-183 in PBMs differentiation is still unknown, and the regulatory relationship between miR-183 and *CFL2* has no reports. This research found that miR-183 may suppress PBM differentiation and confirmed the targeting site with *CFL2*, further clarifying the negative regulation mechanism on the *CFL2* gene. These findings enriched the academic theories of miR-183 on meat tenderness [21] and filled the vacancy of miR-183/CFL2 axis in the field of PBM differentiation.

The quality of meat products was mainly affected by the muscle structure and histological characteristics of myofiber, which closely related to muscle cell differentiation [35]. In this experiment, we demonstrated that miR-183 restrained myoblast differentiation and preliminarily confirmed the profile trend of the *CFL2* gene and related marker genes’ effects on myoblast differentiation during one week. Muscle differentiation is a complex process that involves a multistage gene expression and regulation [36,37,38]. Many research results have reported that CFL2 intensifies C2C12 cell line differentiation. In C2C12 cells, knock-down CFL2 has given rise to F-actin accumulation, accelerated cell cycle progress and increased cell proliferation [11]. Previous studies showed that miR-325-3p mimic suppressed the expression of CFL2 but elevated the accumulation of F-actin, induced the translocation of nuclear YAP, delayed myogenic differentiation, accelerated the progression of cell cycle and ultimately promoted myoblast proliferation [21]. In this experiment, the expression level of *CFL2* increased gradually during C2C12 differentiation, the peak of related marker genes occurred at different time nodes; then, all showed a downward trend after DM6 introduction. Due to the fact that during myogenesis, myoblasts differentiate into myocytes and myotubes exhibit an opposite relationship, when myotubes were completely formed, myoblasts have been fully differentiated, and we inferred that marker genes *MYH3*, *MYOG* and *MYOD* were activated at different myoblast differentiation phases, revealing that *CFL2* may be essential for myoblast differentiation.

Earlier work has authenticated an increase in *CFL2* mRNA during skeletal muscle myogenesis of mice [39]. Thus, we further confirmed the specific regulatory function of *CFL2* in myoblast differentiation by transfected recombinant adenovirus vectors into PBMs. The results of the present study exhibited a significantly higher expression of *CFL2* in the late differentiation stage (5–7 d, myotubes to form myofiber) than in the initial differentiation stage (1–3 d, mononucleated fuse to multinucleated myotubes). Subsequently, *CFL2* gene overexpression had obviously increased the mRNA level of *MYH3*, *MYOG* and *MYOD* and knock-down *CFL2* gene obviously decreased the mRNA levels of *MYH3*, *MYOG* and *MYOD* during PBM differentiation. This means that *CFL2* tends to promote myoblasts differentiation. Recent evidence indicates that cofilin family genes (*CFL1* and *CFL2*) participate in actin cytoskeleton remodeling [40]. Notably, the *CFL1* and *CFL2* genes can be expressed simultaneously in embryonic skeletal muscle, whereas the *CFL2* gene replaces the *CFL1* gene in the late stage of differentiation [6,8]. This means that cofilin only presents one subtype (*CFL1* or *CFL2*) along with the muscle cell growth and development in healthy mouse [41]. This fact may be related to the myogenic regulatory roles of the *CFL2* gene in myoblasts; therefore, we intend to develop a new system for myogenic differentiation monitored by CFL2 from the perspective of molecular genetics and epigenetics. CFL2 depletion inactivated cell differentiation progression and inhibited the expressions of myogenic transcription factors, thereby eventually leading to impaired myogenic differentiation in myoblasts [11,42]. In light of this, we observed that decreased expression of *CFL2* significantly suppressed myoblast differentiation. Our research results were consistent with previous results and showed that CFL2 promoted myoblast differentiation, which is also sufficient for the induction of muscle differentiation.

## 5. Conclusions

In summary, our study reveals that CFL2 was predominantly expressed in adult muscle tissues. It also promoted PBM differentiation by negatively targeting epigenetic modification in cattle. Our results will provide benefits for ameliorating the cattle meat production and meat quality.

## Figures and Tables

**Figure 1 bioengineering-09-00729-f001:**
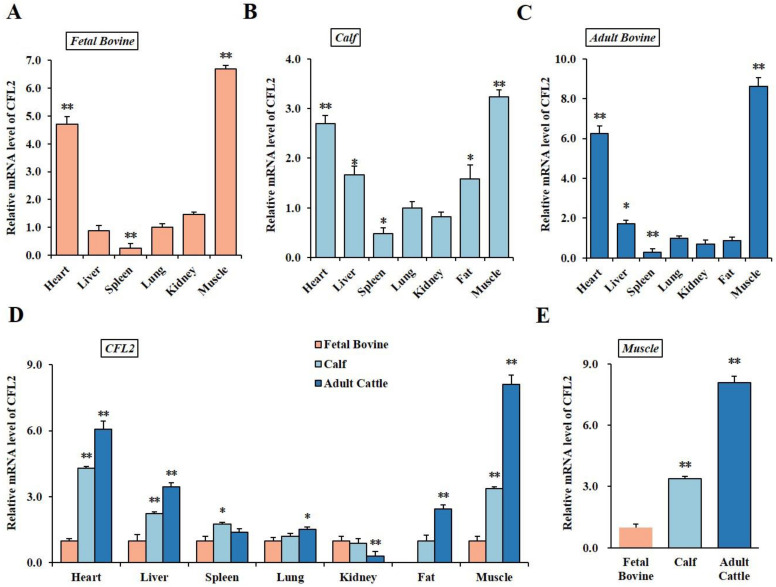
Spatiotemporal expression profiles of *CFL2* gene in cattle tissues. qRT-PCR was used to examine *CFL2* gene mRNA levels of spatiality in six tissues at FB stage (**A**), seven tissues at calf stage (**B**), seven tissues at AC stage (**C**); *CFL2* mRNA levels in lung tissues were taken as 1. qRT-PCR was used to investigate *CFL2* gene mRNA levels of temporally at three developing stages along with cattle age (**D**,**E**); *CFL2* mRNA levels at FB stage were taken as 1. Due to the no fat deposition in FB stage, *CFL2* mRNA levels at calf stage were taken as 1 in adipose tissues. Mean data ± SE of n = 3 independent experiments, each performed in triplicate, and normalized to *GAPDH*. * *p* < 0.05 and ** *p* < 0.01.

**Figure 2 bioengineering-09-00729-f002:**
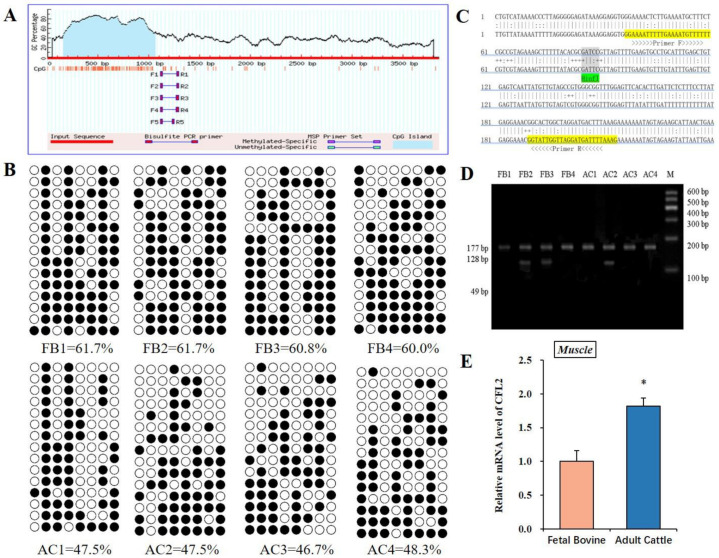
DNA methylation analysis of CFL2 promoter region in muscle tissues. Schematic representation of the proximal promoter region (+1 to −3850 base pairs) to predict high-GC-content regions of *CFL2* gene. In *CFL2* gene promoter region, there were 177 bp *CFL2* DMR (**A**). BSP assay was used to analyze DNA methylation profiles in FB and AC muscle tissues. There are few rows, one for each of the monoclonal bacterial antibodies; there are 8 columns, meaning 8 CpG islands; black circles, meaning methylated CpG region; and white circles, meaning unmethylated CpG region (**B**). The 177 bp *CFL2* DMR contains eight CpG islands, lower strands present bisulfite-converted bases, and yellow highlights and arrows indicate primer sequences using *Hin*fI (“GANTC”) to cut 4th–5th CpG islands for COBRA assay (**C**). COBRA assay was performed to analyze the DNA methylation patterns in FB and AC muscle tissues, each performed in quadruplicate. M means Marker I (**D**). We further examined the *CFL2* gene mRNA levels in FB and AC muscle tissues. Mean data ± SE of n = 3 independent experiments, each performed in quadruplicate, and normalized to *GAPDH*, (* *p* < 0.05) (**E**).

**Figure 3 bioengineering-09-00729-f003:**
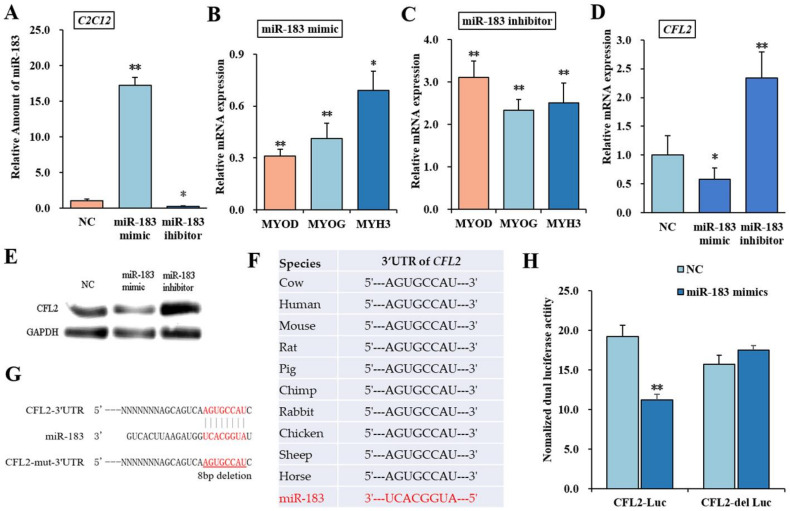
Bta-miR-183 target muscle type CFL2. Expression efficiency of miR-183 was examined after treating bta-miR-183 mimic and inhibitor for 48 h in C2C12 cells (**A**). The expression of myoblast differentiation marker genes *MYOD*, *MYOG*, and *MYH3* were detected by qRT-PCR after transfecting miR-183 mimic or inhibitor into C2C12 cells and inducing for 4 d with 2% horse serum (**B**,**C**). The expression level of miR-183 was normalized to U6, the mRNA levels of *CFL2* and three marker genes were normalized to *GAPDH*. CFL2 mRNA (**D**) and protein (**E**) expression in PBMs were detected by qRT-PCR and Western blot after being transfected with miR-183 mimic, miR-183 inhibitor and NC for 48 h. Putative binding sites of miR-183 on the 3′UTR fragments of *CFL2* mRNA were detected in different species (**F**). Computational prediction of the targeting sites of miR-183 on the 3′UTR fragments of *CFL2* was conducted in Bos Taurus, and 8 base deletion of CFL2 3′UTR was designed for mutant type (**G**). PBMs co-transfected with pairwise plasmids (bta-miR-183 mimic and CFL2-Luc, NC and CFL2-Luc, bta-miR-183 mimic and CFL2-del Luc, NC and CFL2-del Luc), and Renilla luciferase activity was normalized to the firefly luciferase activity (**H**). NC, Negative Control. Mean data ± SE of n = 3 independent experiments, each performed in triplicate. * *p* < 0.05 and ** *p* < 0.01.

**Figure 4 bioengineering-09-00729-f004:**
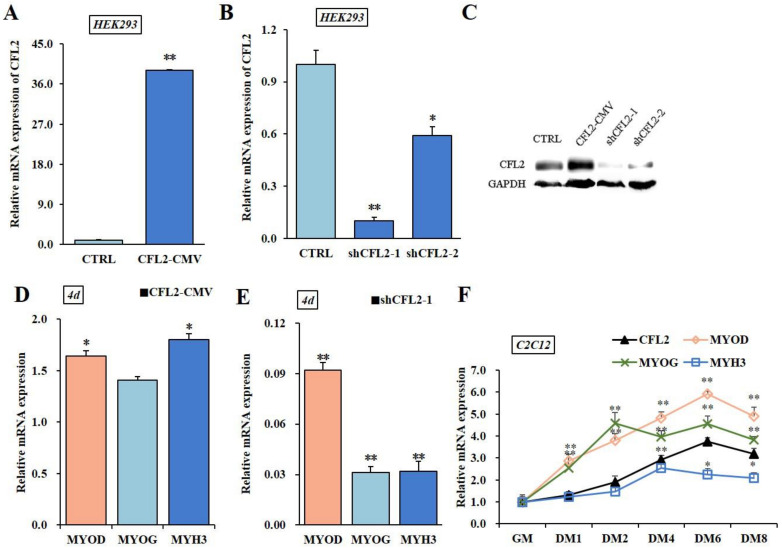
Preliminary functional verification of *CFL2* gene adenovirus vectors. mRNA expression efficiency of *CFL2* was detected after *CFL2* gene overexpression (**A**) or *CFL2* gene interference (**B**) in HEK293 cells. The protein levels of *CFL2* gene were examined by Western blot after being treated with shCFL2-1 or shCFL2-2 for 48 h in HEK293 cells (**C**). The mRNA levels of marker genes were examined after being treated with CFL2-CMV (**D**) or shCFL2-1. (**E**) induced C2C12 cells differentiated for 4 days. C2C12 cells were cultured for 8 days after DM replaced GM, and the mRNA change tendency of *CFL2* gene and three marker genes was further examined by qRT-PCR (**F**). GM means growth medium. DM means differentiation medium. CTRL means Control. Mean data ± SE of n = 3 independent experiments, each performed in triplicate, and normalized to *GAPDH*. * *p* < 0.05 and ** *p* < 0.01.

**Figure 5 bioengineering-09-00729-f005:**
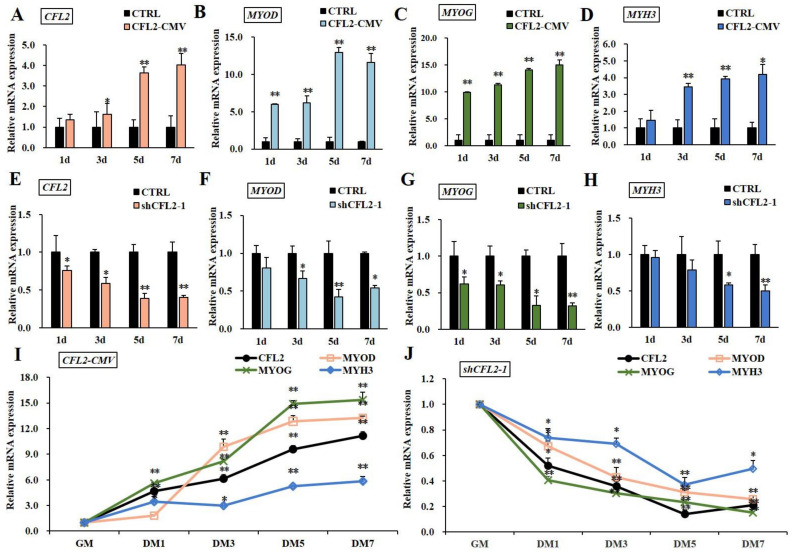
Effects of the *CFL2* gene on the differentiation of PBMs. After being treated with CFL2-CMV (**A**–**D**) or shCFL2-1 (**E**–**H**) induced cells differentiate for 7 days by DM in PBMs; subsequently, the mRNA levels of *CFL2* gene and myogenic marker factors *MYOD*, *MYOG* and *MYH3* were examined once on alternate days. Then, the developmental expression patterns of *CFL2* gene and three myogenic marker genes from GM to DM7 were further examined after being treated with CFL2-CMV (**I**) or shCFL2-1 (**J**) in PBMs. Mean data ± SE of n = 3 independent experiments, each performed in triplicate, and normalized to *GAPDH*. * *p* < 0.05 and ** *p* < 0.01.

**Table 1 bioengineering-09-00729-t001:** Primer information (containing protective bases and restriction sites) for adenovirus vector construction.

Name	Primer Sequences (5′-3′)
CFL2-CMV-F	CGGggtaccATGGCTTCTGGAGTTAC
CFL2-CMV-R	CCCaagcttTCAcatcatcaccatcaccatTAAGGGTTTTCCTTC
shCFL2-1F	gatccCTGAAAGTGCACCGTTAAATTCAAGAGATTTAACGGTGCACTTTCAGtttttta
shCFL2-1R	agcttaaaaaaCTGAAAGTGCACCGTTAAATCTCTTGAATTTAACGGTGCACTTTCAGg
shCFL2-2F	gatccGCTCTAAAGATGCCATTAATTCAAGAGATTAATGGCATCTTTAGAGCtttttta
shCFL2-2R	agcttaaaaaaGCTCTAAAGATGCCATTAATCTCTTGAATTAATGGCATCTTTAGAGCg
shRNA-NC-F	gatccTTCTCCGAACGTGTCACGTTTCAAGAGAACGTGACACGTTCGGAGAAtttttta
shRNA-NC-R	agcttaaaaaaTTCTCCGAACGTGTCACGTTCTCTTGAAACGTGACACGTTCGGAGAAg

**Table 2 bioengineering-09-00729-t002:** Primer information (containing protective bases and restriction sites) for dual luciferase report assay.

Name	Primer Sequences (5′-3′)
CFL2-wild-F	CCGCTCGAGGGAGGCAATGTAGTAGTTTC
CFL2-wild-R	ATAAGAATGCGGCCGCCAAGGCAGGTGAGGTGTATG
CFL2-mut-F	TATAAAGCAGTCAACTGGATCTTAAGGAG
CFL2-mut-R	TCCTTAAGATCCAGTTGACTGCTTTATAAG

**Table 3 bioengineering-09-00729-t003:** Primer information for qRT-PCR.

Name	Primer Sequences (5′-3′)
CFL2-F	GGTGACATTGGTGATACTG
CFL2-R	CATATCGGCAATCATTCAGA
bta-miR-183-RT	GTCGTATCCAGTGCAGGGTCCGAGGTATTCGCACTGGATACGACCAGTGAAT
bta-miR-183-F	ACACTCCAGCTGGGTATGGCACTGGTAGA
miR-R	GCAGGGTCCGAGGTATTC
U6-F	GCTTCGGCAGCACATATACTAAAAT
U6-R	CGCTTCACGAATTTGCGTGTCAT
GAPDH-F	AGATAGCCGTAACTTCTGTGC
GAPDH-R	ACGATGTCCACTTTGCCAG

**Table 4 bioengineering-09-00729-t004:** Primer information of CFL2 DMR for BSP.

Name	Primer Sequence (5′-3′)
CFL2-DMR-F	GGAAAATTTTTGAAAATGTTTTTT
CFL2-DMR-R	CTTTAAAATCATCCTAACCAATACC

## Data Availability

Not applicable.

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
