# Peer review of "New Insight into Muscle-Type Cofilin (CFL2) as an Essential Mediator in Promoting Myogenic Differentiation in Cattle"

_bioengineering, 2022, doi:10.3390/bioengineering9120729_

Round 1

Reviewer 1 Report (Previous Reviewer 3)

The introduction, the results and discussion are clearly presented.

I only have small suggestions for changes:

At the bottom of Figure 1, it should be explained that the fat tissue samples were normalized using calf data and fetus data. It is explained in the text but not in the figure footnote.

In the Figure 3A you are measuring relative amount of mIR-183, but not relative expression of mIR-183

Author Response

Dear Reviewer,

We sincerely appreciate the careful reading of our manuscript entitled “New insight of muscle type cofilin (CFL2) as an essential mediator to promote myogenic differentiation in cattle” (ID: bioengineering-2039564) and your valuable suggestions. Our point-by-point responses to your comments have been provided below, along with a clear indication of the location of the relevant revisions.

  1. Thank you for your comments. I have added the description about fat tissue samples in the Figure 1D footnote.
  2. Thank you for your comments. I have modified the vertical coordinate legend about Figure 3A. Replaced with a new Figure 3.

Thank you for your comments again.

Best regards,

Yujia Sun

Reviewer 2 Report (Previous Reviewer 2)

I regret to say that my concerns were not addressed by the authors. There was perhaps a misunderstanding.

1. the argument: methylation leads to cofilin suppression; cofilin suppression leads to muscle differentiation, therefore methylation leads to muscle differentiation is tenuous, as not DIRECT relationship has been shown. At the very least this should be discussed and qualified

2. with lack of differentiation, I meant showing that muscle cells indeed differentiate (change morphologically, fuse and start expressing muscle proteins). This has not been shown. Either show, or discuss and qualify

Author Response

Dear Reviewer,

We sincerely appreciate your earnest and sincere comments about the manuscript “New insight of muscle type cofilin (CFL2) as an essential mediator to promote myogenic differentiation in cattle” (ID: bioengineering-2039564) and your valuable suggestions.

  1. Thank you for your comments again. I guess I have got the point, it was inappropriate to draw a definite conclusion between DNA methylation and muscle differentiation. The study of DNA methylation modification is a supplement data to temporal expression profiles of previous part research. I have rewritten the conclusion and added part discussion in the revised manuscript. Thank you for your suggestion again.
  2. Thank you for your comments again. There was a misunderstanding in previous comments. Now I will provide the original differentiation pictures of bovine primary myoblast cells (reply in word version) and added in the supplementary data (Figure S3). I also explained in the text. Thank you for your suggestion again.

Best regards,

Yujia Sun

Reviewer 3 Report (Previous Reviewer 1)

Dear authors,

I appreciate the responses from the authors. They have cleared the doubts I had in the first evaluation.

Author Response

Dear Reviewer,

We sincerely appreciate the careful reading of our manuscript entitled “New insight of muscle type cofilin (CFL2) as an essential mediator to promote myogenic differentiation in cattle” (ID: bioengineering-2039564) and your valuable suggestions. Thank you for your previous comments again.

Best regards,

Yujia Sun

This manuscript is a resubmission of an earlier submission. The following is a list of the peer review reports and author responses from that submission.

Round 1

Reviewer 1 Report

Dear authors,

This manuscript addresses the effects of the epigenetic modification on mechanism CFL2 gene in relation to its functions in muscle tissue. The study is very well done, the methods used are appropriate. But I am struggling to find a "strong" relationship between CFL2 expression and its epigenetic modifications and meat quality. The results they arrive at are very solid and well analyzed. But the conclusions, regarding the effect on the quality of the meat, I think they are very comprehensive (How do you demonstrate the direct relationship between these studies and the quality of the meat? What kind of feature do they mention? taste? tenderness? ratio of protein content vs. lipid?, etc.). Every time the authors mention the differentiation times (that is, the cells grown in the differentiation medium), it would be interesting if they showed photos of the cells. I understand that the C2C12 in the differentiation medium, need more days to differentiate than those indicated in this work. The authors say that “After induced to 4-day for cell 228 differentiation, we found that the mRNA level of myoblast differentiation marker genes 229 MYOD, MYOG and MYH3 were downregulated by bta-miR-183 mimic”, but with only 4 days, these genes have changed their basal levels, in significant fashion?

The authors could explain why they only used samples from females?. The authors could explain why they only used samples from females Is there a relationship with hormone levels? estrogens/androgens?

Author Response

Dear Reviewer,

We sincerely appreciate the careful reading of our manuscript entitled “New insight of muscle type cofilin (CFL2) as an essential mediator to promote myogenic differentiation in cattle” (ID: bioengineering-1953193) and your valuable suggestions. Our point-by-point responses to your comments have been provided below, along with a clear indication of the location of the relevant revisions.

  1. Our studies focus on the epigenetic mechanisms of muscle-type gene (CFL2) during skeletal muscle fibers differentiation, and found that epigenetic modification of CFL2 regulates skeletal muscle fibers differentiation directly. Previous studies have indicated that the growth and development of skeletal muscle is the primary factor in agricultural meat production and meat quality,and meat quality is tightly correlated with the histological characteristics of muscle fibers (1-3). So, muscle fibers differentiation could affect meat tenderness and other meat quality traits. Thank you for your comments again.

(1) Pas, M.T.; Keuning, E.; Hulsegge, B.; Hoving-Bolink, A.H.; Evans, G.; Mulder, H.A.J.J.o.A.S. Longissimus muscle transcriptome profiles related to carcass and meat quality traits in fresh meat Pietrain carcasses. 2010, 88, 4044-4055.

(2) Covington, R.C.; Tuma, H.J.; Grant, D.L.; Dayton, A.D. Various chemical and histological characteristics of beef muscle as related to tenderness. Journal of Animal Science 1970, 30, 191-196.

(3) Li, C.B.; Xu, X.L.; Zhou, G.H.; Xu, S.Q.; Zhang, J.B. Effects of carcass maturity on meat quality characteristics of beef semitendinosus muscle for chinese native yellow steers. Animal 2007, 1, 780-786.

  1. In this study, the function of bta-miR-183 mimic or inhibitor was maximized after treated into cells 48~72 hours. C2C12 cells would form myotubes after induced 3~4 days. After treating bta-miR-183 mimic or inhibitor into C2C12 cells, we induced differentiation to the 4-day, at this time, muscle satellite cells differentiate into myotubes, it is enough to observe the expression level change of biomarker genes compared to the uninduced growth state. I will attach the cell differentiation pictures if necessary. Thank you for your comments again.
  2. All female cattle were raised at the same level. Female cattle samples were selected as experimental subjects, mainly because female cattle are often used for breeding, and considering heredity and heritability, female cattle experiments will provide a very strong basis of theoretical research for the subsequent genetic improvement of cattle breeds. Thank you for your comments again.

Best regards,

Yujia Sun

Reviewer 2 Report

The report by Sun et al describes three studies: 1. CFL-2 expression and methylation dependent on age; 2. the effect of miR-183 on CFL-2 expression and differentiation genes and 3. the effect of CFL-2 on expression of differentiation genes. The results are clear and consistent. My two major concerns are, first, that no direct relation of CFL-2 methylation with function and that there is no immediate relation shown between CFL-2 and the expression of differentiation genes. The association can still be due to an effect that affects both CFL-2 and myogenic genes as a result of differentiation level. Second, no differentiation was shown in muscle cells, just expression level of early stage myogenic proteins. The conclusion is therefore not fully supported by the conclusion

many grammar issues

Author Response

Dear Reviewer,

We sincerely appreciate the careful reading of our manuscript entitled “New insight of muscle type cofilin (CFL2) as an essential mediator to promote myogenic differentiation in cattle” (ID: bioengineering-1953193) and your valuable suggestions. Our point-by-point responses to your comments have been provided below, along with a clear indication of the location of the relevant revisions.

  1. Thank you for your comments. DNA methylation modification as one of epigenetic. We mainly explored the change of DNA methylation level on myogenesis and muscle maturation. In cells, methyl moieties may directly regulate the action of transcription factors, we further studied the effect of DNA methylation modification on mRNA expression levels, and found that DNA methylation participate in suppression patterns during muscle growth and development through regulating gene expression. There was a negative correlation between DNA methylation status and mRNA expression level. The study of DNA methylation modification is a supplement data to temporal expression profiles of previous part research. I have rewritten the conclusion in the revised manuscript.
  2. I have polished the manuscript language by Research Square agency (AJE department). Thank you for your suggestion again.

Best regards,

Yujia Sun

Reviewer 3 Report

The authors present the CLF2 gene as an important element to promote myogenic differentiation in cattle. The introduction, the results and the discussion are presented correctly, and the contributions are relevant to the knowledge of muscle formation in cattles.

There are some typing errors like, ´´ privious ´´ instead of ´´previous ´´ at line 40 or  ´´has´´ instead of ´´have´´ at line 75; and some uncomplete references as 7, 27, 32….

Author Response

Dear Reviewer,

We sincerely appreciate the careful reading of our manuscript entitled “New insight of muscle type cofilin (CFL2) as an essential mediator to promote myogenic differentiation in cattle” (ID: bioengineering-1953193) and your valuable suggestions. Our responses to your comments have been provided in the revised version of manuscript. I have already checked the typing errors and rewritten some sentences throughout the manuscript. About incomplete information of references, I have added the incomplete information of ref. 7, 26, 35, 39, 45, 46…et.al., whereas ref. 27 was only published online, I could only provide the DOI number. Thank you for your comments again.

Best regards,

Yujia Sun
